# Factors influencing active tuberculosis case-finding policy development and implementation: a scoping review

Olivia Biermann ![ORCID] ,[1] Knut Lönnroth,[1] Maxine Caws,[2,3] Kerri Viney[1,4]

¹Department of Global Public Health, Karolinska Institutet, Stockholm, Sweden
²Department of Clinical Sciences, Liverpool School of Tropical Medicine, Liverpool, UK
³Birat Nepal Medical Trust, Kathmandu, Lazimpat, Nepal
⁴Research School of Population Health, College of Health and Medicine, Australian National University, Canberra, Australian Capital Territory, Australia

**Correspondence to**
Olivia Biermann;
olivia.biermann@ki.se

## ABSTRACT

**Objective** To explore antecedents, components and influencing factors on active case-finding (ACF) policy development and implementation.

**Design** Scoping review, searching MEDLINE, Web of Science, the Cochrane Database of Systematic Reviews and the World Health Organization (WHO) Library from January 1968 to January 2018. We excluded studies focusing on latent tuberculosis (TB) infection, passive case-finding, childhood TB and studies about effectiveness, yield, accuracy and impact without descriptions of how this evidence has/could influence ACF policy or implementation. We included any type of study written in English, and conducted frequency and thematic analyses.

**Results** Seventy-three articles fulfilled our eligibility criteria. Most (67%) were published after 2010. The studies were conducted in all WHO regions, but primarily in Africa (22%), Europe (23%) and the Western-Pacific region (12%). Forty-one percent of the studies were classified as quantitative, followed by reviews (22%) and qualitative studies (12%). Most articles focused on ACF for tuberculosis contacts (25%) or migrants (32%). Fourteen percent of the articles described community-based screening of high-risk populations. Fifty-nine percent of studies reported influencing factors for ACF implementation; mostly linked to the health system (eg, resources) and the community/individual (eg, social determinants of health). Only two articles highlighted factors influencing ACF policy development (eg, politics). Six articles described WHO's ACF-related recommendations as important antecedent for ACF. Key components of successful ACF implementation include health system capacity, mechanisms for integration, education and collaboration for ACF.

**Conclusion** We identified some main themes regarding the antecedents, components and influencing factors for ACF policy development and implementation. While we know much about facilitators and barriers for ACF policy implementation, we know less about *how* to strengthen those facilitators and *how* to overcome those barriers. A major knowledge gap remains when it comes to understanding which contextual factors influence ACF policy development. Research is required to understand, inform and improve ACF policy development and implementation.

## BACKGROUND

Systematic screening for tuberculosis (TB) is defined by the World Health Organization

(WHO) as the 'systematic identification of people with suspected active TB, in a predetermined target group, using tests, examinations or other procedures that can be applied rapidly'.[1] Active case-finding (ACF) is synonymous with systematic screening for active TB, although it usually implies screening outside of health facilities. TB is a major global health challenge, particularly in low-income and middle-income countries. The WHO End TB Strategy and the Sustainable Development Goals aim at finding the 'missing cases' and ending the global TB epidemic by 2030. This will require intensified activity to increase TB case detection, specifically in 'hard-to-reach' groups.[2] In 2019, there was a 3 million gap between estimated incident and notified TB cases globally, reflecting a combination of an underdiagnosis of cases and underreporting of cases who have been detected.[3] Many people with TB are diagnosed only

after long delays,[4] causing suffering and economic hardship for TB patients and TB-affected households, and sustained transmission.[1]

ACF is mostly provider-initiated. It targets people in high-risk groups who may not seek healthcare actively.[1] Potential benefits for patients include reduced morbidity, mortality and economic consequences due to earlier diagnosis, while society can benefit from reduced transmission and a reduced burden of TB, which often affects the most economically productive members of a society. TB screening in high-risk groups has been implemented in many settings within the TB programme or in the context of research and can significantly improve TB case notification.[5–7]

However, if not well targeted and implemented, ACF can be costly leading to diversion and waste of scarce resources, potentially compromising passive case-finding infrastructure and weakening health systems. It can also cause harm to individuals, for example, by increasing the risk of a false positive diagnosis and providing TB treatment to individuals without TB, or increased stigma and discrimination arising from uncovering TB, which need to be weighed against the benefit of identifying cases in the community that would otherwise go untreated.[1 2]

Questions remain about both *if* ACF in general is worthwhile, and *how* to best plan and implement outreach screening through ACF in a given context as a synergistic way, that is, integrated into the healthcare system, and not delivered separately in a parallel system. The evidence base is weak concerning the benefits and cost-effectiveness of ACF and how it varies between risk groups.[8] The strength of evidence to support alternative ACF approaches is low, with few head-to-head trial results being available to inform policy and practice. Many differences exist in stakeholders' values and preferences concerning the rationale, outcomes and possible secondary effects of ACF. Despite the relatively weak evidence, WHO has a guideline on systematic TB screening and the Global Fund[9] and TB REACH[10] provide increasing funding for ACF, while a growing number of countries are increasing investment in and have national policies for ACF. For example, Vietnam's National Strategic Plan 2015–2020 includes ACF in remote and congregate settings, and in high-risk groups including contacts, prisoners, miners, the elderly, homeless people, young adults, migrants, factory workers, communities, people living with HIV, multidrug-resistant patients, young males, smokers and intravenous drug users.[11]

The aim of this study was to explore antecedents, components and influencing factors for ACF policy development and implementation, to inform and improve future ACF policy processes. We did not aim to review the evidence on ACF per se; therefore, we explore yield, accuracy and impact only in the context of how they influence ACF policy development or implementation.

## METHODS

We conducted a scoping review, using an a-priori protocol based on the following research question: *Which are the antecedents, components and influencing factors (barriers and facilitators) in developing and implementing ACF policies?* The scoping review methodology was deemed appropriate given the breadth of the question. The protocol was based on guidance from the Joanna Briggs Methods Manual for Scoping Reviews.[12] The inclusion criteria, the screening manual and the data-charting table were developed by one author (OB), reviewed by two authors (KV and KL) to ensure face validity, and pilot-tested. All team members were consulted at various stages of the scoping review to provide input on the search, data extraction and charting, and the interpretation of the results. The charted data were summarised in tables, and then condensed resulting in tables 1 and 2. The Preferred Reporting Items for Systematic reviews and Meta-Analyses Extension for Scoping Reviews (PRISMA-ScR) was used to guide reporting.[13] The completed checklist is available in online additional file 1. The protocol is available from the corresponding author on request.

### Data sources and search

We searched MEDLINE, Web of Science, The Cochrane Database of Systematic Reviews and the WHO Library from January 1968 (year of the first publication on ACF in MEDLINE) until January 2018. The search strategy was developed in collaboration with a medical librarian from Karolinska Institutet (Carl Gornitzki), and further refined through team discussion. The search strategy for MEDLINE is available in online additional file 2. The latter was standardised but adapted to fit the other database searches. The respective search strategies are available from the corresponding author on request. The results of the literature search were imported into Rayyan.

### Inclusion criteria

Eligible studies focused on the detection of active TB disease. The review included studies describing or analysing ACF policy development and implementation. Facilitators and barriers linked to access or treatment were included, if there was a clear link to ACF. Furthermore, the review included studies analysing the use of evidence in ACF policy development and implementation. Studies of any design, conducted in any setting or country and those published in the English language were eligible for inclusion. We excluded studies focusing only on latent TB infection, passive case-finding and childhood TB, and studies about effectiveness, yield, accuracy and impact without descriptions of how this evidence has/could influence ACF policy. The inclusion of different ACF approaches implies that while many lessons can be learnt across different settings, some might not be generalisable.

### Screening, data abstraction and charting

One reviewer (OB) initially reviewed titles to remove duplicates, and those studies not focusing on TB. Two

**Table 1** Characteristics of the articles included in the scoping review (n=73)

| Study characteristics | References | Number (%) |
|---|---|---|
| **Year of publication** | | |
| 1979 | 66 | 1 (1) |
| 1995–1999 | 51 53 58 67 | 4 (5) |
| 2000–2009 | 17 21 23 24 28 30 31 46–48 54 56 59 68–70 | 16 (22) |
| 2010–2018 | 14–16 18–20 22 25–27 29 32 33 35–42 44 45 49 50 52 55 57 60–65 71–83 85 | 49 (67) |
| **Region** | | |
| Africa | 14 20 22 35 41 42 44 45 60 62 63 65 66 73 82 85 | 16 (22) |
| Americas (North America) | 23 24 32 34 58 61 64 67 70 72 | 10 (14) |
| Eastern Mediterranean | 77 | 1 (1) |
| Europe | 28 30 38 43 46–48 52–55 59 68 69 81 83 84 | 17 (23) |
| South-East Asia | 50 51 63 80 86 | 5 (7) |
| Western Pacific | 15 16 31 36 39 40 63 75 76 | 9 (12) |
| Global perspective | 19 21 27 33 74 76 78 | 7 (10) |
| Other* | 17 29 49 71 | 4 (5) |
| **Study design** | | |
| Quantitative | 15 16 29–31 35–37 40 41 44 45 48–51 59 61 63–68 73 75 77 83 84 86 | 30 (41) |
| Review | 17–20 25 27 34 46 49 56 69 70 74 76 78 79 | 16 (22) |
| Qualitative | 14 22 23 38 43 54 55 62 71 | 9 (12) |
| Descriptive | 28 32 71 81 85 | 5 (7) |
| Case study | 26 58 72 | 3 (4) |
| Mixed methods | 39 42 | 2 (3) |
| Other† | 33 47 52 53 | 4 (5) |
| **Target group** | | |
| Contacts | 14 15 21 22 24 27 32 35–37 42 44 74–76 81 82 84 | 18 (25) |
| Migrants | 20 28 30 34 38 43 46–49 54–56 59 61 64 68 69 71 73 77 79 83 | 23 (32) |
| Community | 17 26 29 31 40 45 51 60 62 63 | 10 (14) |
| High risk groups | 19 25 33 57 66 | 5 (7) |
| Homeless | 23 53 58 67 | 4 (5) |
| Asylum seekers or refugees | 50 52 | 2 (3) |
| People living with HIV | 18 78 | 2 (3) |
| Urban poor | 16 39 | 2 (3) |
| Other‡ | 41 70 72 80 85 86 | 6 (8) |

*High HIV prevalence countries,[17] high TB incidence countries,[29] low-burden countries.[49 71]
†Letter to editor,[52] observational study,[53] perspective[33] and editorial.[47]
‡Population affected by TB outbreak,[72] pastoralists,[41] healthcare workers,[65] prisoners,[80] people living in slums,[86] street connected youth and young adults[85] and non-immigrant visitors.[70]

reviewers (OB and KV) independently reviewed titles and abstracts, and then full-text articles for inclusion. Conflicts were resolved through discussion based on the inclusion/exclusion criteria.

The data-charting table was developed in Microsoft Excel. An abridged version of the table is available in online additional file 3. Data were extracted on study design, country, target population for ACF, benefits of ACF, risks of ACF, ACF antecedents, ACF policy development and implementation (use of evidence, country-level to individual-level facilitators and barriers, stakeholders involved), lessons learnt, future perspectives and future research. Relevant data were charted by one author (OB), while a second author (KV) verified this step by charting data from a random sample of studies for comparison. Differences in data-charting were resolved by discussion.

**Patient and public involvement**

This study is related to a qualitative study of experts and a survey of National TB Programme managers combined with a policy document review on the topic of ACF policy development and implementation which are currently under way to harness tacit knowledge on the topic. The research questions and variables for these studies were

**Table 2** Barriers and facilitators for active case-finding policy implementation (n=43)

| Reported factors | References | Number (%) |
|---|---|---|
| **Health system level** | | |
| Limited financial resources | 16 21 22 26 27 32 39 40 43 46 48 51 54 66 78 | 15 (35) |
| Existing systems and structures | 14 19 21 27 38 43 44 53 54 69 78 | 11 (26) |
| Availability of diagnostic tests and services | 16 19 27 34 41 49 50 66 | 8 (19) |
| Staff experience, expertise, motivation | 14 17 24 26 27 39–41 43 46 52 55 58 71 | 14 (33) |
| Over-burdening staff | 16 21 26 39 41 51 70 | 7 (16) |
| Using person-centred approaches | 14 22 23 27 34 43 53 65 | 8 (19) |
| Collaboration and integration | 14 16 18 34 39 43 52 | 7 (16) |
| Health education | 34 40–43 | 5 (12) |
| **Health information and research system level** | | |
| Data, systems, supervision | 19 24 27–29 42 48 | 7 (16) |
| **Individual and community level** | | |
| Stigma and discrimination | 14 22–25 27 28 31 37 38 43 50 58 59 77 82 | 16 (37) |
| Individual characteristics | 23 31 34 37 39 43 51 58 65 66 77 83 | 12 (28) |
| Sociocultural factors | 24 25 27 28 34 41 43 46 47 54 56 59 77 83 | 14 (33) |
| Fear | 23 24 34 38 39 43 53–56 | 10 (23) |
| Mistrust | 14 24 41 43 46 47 51 53 57 72 | 10 (23) |
| Knowledge and awareness | 14 23 24 26 37 43 53 58 | 8 (19) |

informed by the results of this scoping review. The results of this study will be presented to researchers and policy-makers in the field via targeted issue briefs. We will also share the results with the public via a video and short messages on social media.

## RESULTS

### Literature search

After screening 2943 titles and abstracts and 271 full-text articles, 73 unique articles fulfilled our eligibility criteria (figure 1).[14–86] The reasons for excluding full-text articles are provided in figure 1. What all included studies have in common is the goal of finding 'missing' TB cases through ACF, while the approaches vary greatly; from contact investigation to screening people in home-less shelters (table 1). An example of a study included in this review is a qualitative study by Ayakaka *et al*, which explored influencing factors for TB contact investigation in Kampala, Uganda.[14] The stakeholders who were inter-viewed described key barriers for ACF as being limited knowledge about TB among contacts, stigma, mistrust of health centre staff among index patients and contacts, and high travel costs for lay health workers and contacts. At the same time, key facilitators for ACF comprised personalised and enabling services provided by lay health workers. To overcome barriers and strengthen facilita-tors, the researchers identified education, incentivisation and restructuring of the service environment as relevant interventions.[14]

### Characteristics of the included articles (n=73)

The breadth of studies identified reflect the complexity and diversity of ACF policies and their characteristics. There has been a gradual increase in the number of arti-cles on ACF since the first identified (eg, from 1979), and 67% (n=49/73) were published in the period 2010–2018 (table 1). The studies were conducted in all WHO regions, that is, the African region (22%, n=16/73), the Americas (North America) (14%, n=10/73), the Eastern Mediterranean region (1%, n=1/73), the European region (23%, n=17/73), South-East Asia (7%, n=5/73) and the Western Pacific region (12%, n=9/73). The most common types of articles were classified as quantitative papers (41%, n=30/73), reviews (22%, n=16/73) and qualitative studies (12%, n=9/73). The most frequent ACF target groups were contacts (25%, n=18/73) and migrants from high-incidence countries (32%, n=23/73), while 14% percent (n=10/73) of the articles focused on screening of defined communities. The remaining studies focused on different target groups which are specified in table 1.

### Antecedents

ACF has been implemented for many decades primarily in high-income countries, starting with mass screening campaigns in the general population in the 1950s and 1960s, then moving towards specific risk populations in recent decades, such as migrants from high-incidence countries and prison populations.[15] In low-income and middle-income countries, the interest in ACF has increased in recent years, mainly as a response to a sustained case detection gap documented in annual

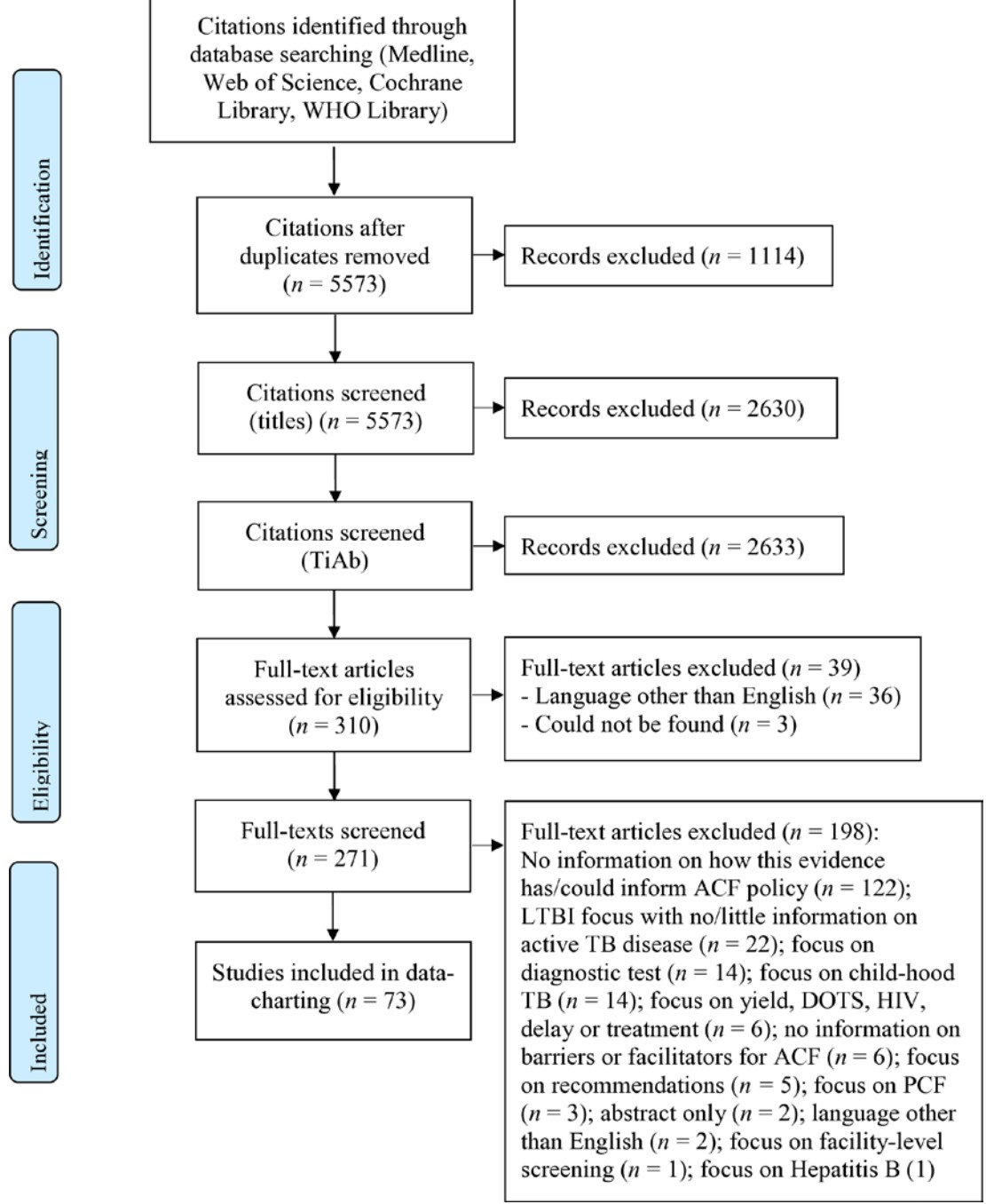

**Figure 1** PRISMA flow diagram. ACF, active case-finding; PRISMA, Preferred Reporting Items for Systematic Reviews and Meta-Analyses; TB, tuberculosis.

Global TB Reports produced by WHO[3] and the emergence of new WHO guidelines.[16]

Generally, TB programmes have moved from traditionally vertical approaches and exclusively facility-based case-finding among people seeking care with TB symptoms (so called 'passive case-finding') towards being closer to the client through decentralised, community-based solutions and outreach activities, including ACF—similar to community-based testing services that have increased access to HIV testing and care.[17 18]

In 1976, the WHO Expert Committee on TB recommended that countries abandon *indiscriminate* mobile mass radiography, as evidence showed the inefficiency of population-wide screening in settings that had seen TB rates drop dramatically since the World War II.[19] It should be noted that the Expert Committee emphasised that TB screening should still be done in selected risk groups.[19 20]

The current WHO guidelines on systematic screening for active TB were released in 2013.[1] These guidelines still discourage *indiscriminate* mass screening and strongly recommend ACF only in selected high-risk groups, while conditionally recommending screening in other high-risk groups. Conditional recommendations are those for which the benefits are expected to outweigh the risks,

while the trade-offs, such as cost-effectiveness, feasibility or affordability may be uncertain. For example, WHO has made a conditional recommendation (very low-quality evidence) for screening certain clinical risk groups in settings where the TB prevalence is over 100 per 100 000 population (while not recommending such screening in countries with a prevalence of less than 100 per 100 000 population).[1] The 2013 WHO guidelines—reflected in the latest 2015 WHO's End TB Strategy[2] and aligned with the Sustainable Development Goals—put ACF back on the agenda. The guidelines provide principles for how to test and expand an approach that had been largely missing in previous global TB strategies, including the Directly Observed Treatment, Short Course Strategy from 1994[87] and the Stop TB Strategy 2006–2015 (although contact investigation and screening people living with HIV was included in the latter).[88] Moreover, the guidelines emphasise the importance of basing ACF policy on local epidemiology and to perform careful evaluations of ACF implementation, yield and impact to improve the ACF evidence base.[20] Today, ACF is considered a global health priority.[21 22]

## Components

According to WHO,[1] the main components of ACF include (1) having high-quality TB diagnosis, treatment, care, management and support for patients, as well as the capacity to scale these up (if needed), before screening is initiated; (2) prioritising risk groups; (3) choosing a suitable screening algorithm; (4) following established ethical principles; (5) optimising synergies with the delivery of other health and social services and (6) monitoring and reassessing ACF approaches.

The components of ACF, as well as those of the wider health system that ACF depends on, are well described in the literature. It is important for high-quality diagnosis, treatment, care, management, patient support and capacity for scale-up to be available.[19] Risk groups must be prioritised based on assessments of benefits and risks, feasibility, acceptability, the number needed to screen to detect a case of TB and the cost-effectiveness of ACF,[19] while also considering the heterogeneity of the risk group.[23]

The overall ACF approach must be flexible, patient-centred[15 24] and culturally sensitive.[25] In resource poor settings, ACF is often backed by multichannel financing mechanisms, including from international donors.[26] Record-keeping, monitoring and evaluation are described as being essential.[27 28] Incentives are sometimes deemed necessary, for example, for health workers,[5 29] laboratory technicians,[30] screening participants[11 31–35] and TB patients.[29 35 36] The articles also emphasised ACF-specific health education as paramount—for patients,[31 37–39] their families,[40] providers[24] and community health workers (CHW).[36 37] The training of CHWs should be followed by supervision.[41 42] In addition, community involvement[32 39 42 43] and social mobilisation[29] are major parts of ACF. One study described how the involvement of community leaders was a key component to improve trust among patients and contacts.[31]

Finally, the studies underlined how vital it is to collaborate with other disease programmes,[33 34 43] agencies and sectors[26 27] and to integrate ACF within existing systems.[18 44] The latter includes collaboration between providers and community-based organisations,[43] and between HIV and TB services.[45]

## Influencing factors

In the following section, we summarise influencing factors for ACF policy development and implementation. We chose to speak of 'influencing factors' rather than facilitators and barriers, as one and the same factor might be a facilitator in one context and a barrier in another, for example, health workers' satisfaction might be a powerful facilitator for ACF,[42] while the lack thereof could be a strong barrier as well.[46]

## Influencing factors for ACF policy development

ACF policy development processes were not well described in the literature. Policy development comprises the investment of governments and donors in ACF. Only two articles reported facilitators and/or barriers for ACF policy development: politics[46 47] and laws.[47] Welshman[46] describes how, in the mid-1950s, the Ministry of Health in the United Kingdom subverted pressure from stakeholders who were in favour of compulsory ACF at ports of entry—the Ministry 'argued that the problem (of TB in migrants) was a minor one' so a policy was not developed. On the other hand, laws such as the 1958 Commonwealth Migration Act were described, which made screening compulsory, and thus influenced ACF policy development in the UK.[47]

## Influencing factors for ACF policy implementation

Policy implementation processes in the area of ACF are widely researched. In total, 43 articles reported barriers and/or facilitators for ACF implementation (table 2). Most articles mentioned factors at the level of the health system, as well as the individual and community level.

### Health system level

It is fundamental to consider health system factors when putting ACF policy into practice and identifying which factors influence ACF policy implementation the most. We derived six major themes from the articles: (1) availability of financial resources, (2) existing systems and structures, (3) availability of diagnostic tests, (4) staff experience and motivation, (5) collaboration between different actors and (6) implementation of a person-centred approach.

### Availability of financial resources

Many articles (35%, n=15/43) mentioned financial resources as an influencing factor for ACF policy implementation. Finances were described as being scarce at different levels of the health system,[40] for the provision of staff,[39 42] drugs and equipment.[48]

### Existing systems and structures

Existing systems and structures were reported as influencing factors for ACF policy implementation by 26% (n=11/43) of the articles. Studies elaborated on the availability of health and social services[14 19] and laboratory networks[21] that could be used for ACF implementation.

### Availability of diagnostic tests of sufficient sensitivity and specificity

Nineteen percent (n=8/43) of the articles elaborated on the availability of diagnostic tests and services as key influencing factor for ACF policy implementation—with many articles referring to the availability of molecular diagnostic tests such as Xpert MTB/RIF.[27 36 49 50] Decentralisation of healthcare services, national coverage of diagnostics and task sharing were perceived as facilitators of ACF policy implementation, but unavailable in many settings.[33]

### Staff experience and motivation

Thirty-three percent (n=14/43) of the studies described staff experience and motivation as influencing factors for ACF policy implementation. One study elaborated on how, on the one hand, staff members would feel motivated by increased case detection and awareness-raising through ACF. On the other hand, they would be frustrated when being 'shouted at, ignored, (and) harassed' by target communities, in this case people living in poor urban settlements.[39] The over-burdening of staff was highlighted by 16% (n=7/43) of the articles, for example, leading to absenteeism,[51] and thus inhibiting ACF implementation.

### Collaboration between different actors

A variety of studies (16%, n=7/43) described collaboration as an important influencing factor for ACF policy implementation. One example is the collaboration between community workers, health centre staff and laboratory technicians in delivering diagnostic services as part of ACF.[16] Moreover, Akkerman *et al* described the positive impact on diagnostic accuracy and the proportion of follow-up medical interventions when there is close collaboration with a limited number of experienced, high-volume chest radiograph readers when screening asylum seekers and other high-risk groups.[52]

### Implementation of a person-centred approach

The need for a person-centred approach to ACF policy implementation was referred to by 19% (n=8/43) of the articles, for example, adopting approaches including healthcare providers being non-judgemental and supportive of patients and their families during diagnosis and follow-up,[53] and assuring confidentiality.[22 52] Another 12% (n=5/43) of the articles emphasised health education for the general public, screening participants, as well as TB patients and their families as a necessity.

### Individual and community

ACF policies should be *for* individuals and communities and their influence on successful implementation is elaborately described in the literature. Three overarching themes emerged from the studies: (1) stigma and discrimination, (2) individual characteristics and sociocultural factors and (3) knowledge and awareness.

### Stigma and discrimination

Many studies (37%, n=16/43) reported stigma and discrimination linked to TB as influencing factors for ACF policy implementation. The consequences could be the avoidant behaviour of a contact of an index TB case, such as providing incorrect phone numbers and/or wrong directions to their homes,[14] or misconceptions about TB.[24 43] Issues of fear, either of TB disease,[23 39 53] the process of ACF,[54] or the wider implications of being identified as a TB case, for example, through migrant screening programmes, where being diagnosed with TB case has in some instances resulted in deportation.[34 38 55 56] Studies also reported the lack of trust of patients and their families as key barrier for ACF policy implementation[41 51]—not only for early diagnosis, but for the sustained commitment to addressing social determinants of TB.[57]

### Individual characteristics and sociocultural factors

Twenty-eight percent (n=12/43) of the articles mentioned the social characteristics of the persons screened as key influencing factors for ACF policy implementation, for example, for ensuring equitable access to care[21 39 58] and promoting conducive health-seeking behaviour.[23 41 43] Moreover, the articles frequently reported sociocultural factors and language (33%, n=14/43) as influencing factors for ACF policy implementation. Culture was commonly described as a barrier to ACF policy implementation,[34 41 59] for example, if cultural beliefs would negatively influence the receptiveness to community health workers' advice. Only one study mentioned culture as a facilitator for putting ACF into practice, for example, if culture meant respect towards professional authority.[59]

### Knowledge and awareness

Finally, 19% (n=8/43) of the articles reported factors related to knowledge and awareness as influencing factors for ACF policy implementation. One such factor could be the understanding of the purpose of ACF.[24] Also, knowledge and awareness are closely linked to persisting stigma and discrimination, for example, Ayakaka *et al*[14] describe how health workers attributed TB-associated stigma to a lack of general knowledge in the community about TB.

## DISCUSSION
### Characteristics of the included articles

We identified 73 articles addressing antecedents, components and/or influencing factors for ACF policy development and implementation. The included studies originated from all WHO regions. Africa and Asia are

home to 90% of the 30 high TB burden countries (53% and 37%, respectively),[89] which may explain why studies were more frequently conducted in these continents: 22% of the studies were conducted in Africa, 12% in the Western-Pacific region and 7% in South-East Asia. Studies conducted in the European region (23%) were predominantly about different forms of migrant screening. There was no study from Latin America among the 73 included articles, and there were no articles in the Spanish language among those excluded.

Most studies were published after 2010, which may be a result of programmes implementing WHO guidelines on systematic screening[1] as well as the growing number of prevalence surveys in different settings.[39] We classified 41% of the articles as quantitative, for example, cross-sectional and cohort studies, effectiveness and cost-effectiveness studies, randomised controlled trials and programme evaluations. The next most common study types were reviews (22%) and qualitative studies (12%). The first qualitative studies on our topic were published in 2003.[24 54] We only identified two mixed-methods studies, one published in 2013[42] and one in 2015.[39] The mixed-methods design combined with quantitative ACF evaluation might become more commonly used to explore the complexity of ACF policy development and implementation in the future, as it has the potential to both increase contextual understanding and reduce biases.

Among the broad range of possible target groups for ACF, most of the articles focused either on TB contacts or migrants. The focus on the latter was mainly in low-incidence countries and may be a result of long-standing national health and migration policies supporting these types of screening. Yet, other studies covered vulnerable groups, such as homeless people, people living with HIV or people living in congregate settings, as well as the urban poor. Some articles included analysis of more than one high-risk group. There were no studies on indiscriminate mass screening. However, there were studies targeting broad communities, for example, communities with high HIV prevalence.

### Antecedents

Despite the growing number of studies (eg, operational studies), there is still a relative lack of evidence about the health benefits for individuals, the epidemiological impact and cost-effectiveness of different ACF approaches. Nonetheless, the interest in ACF has clearly increased in recent years. The persistent case detection gap, the 2013 WHO guidelines on systematic screening[1] and ACF being mentioned as a core component of the End TB Strategy[2] seem to have motivated countries to include ACF in national TB strategies. Universal Health Coverage and the Sustainable Development Goals have only marginally been mentioned as important for stepping up the TB response.[33] However, the Sustainable Development Goals—which aim to 'leave no one behind'—might have contributed to a greater interest in ACF to leave no undiagnosed TB patients behind.

Not having enough evidence to drive ACF policy development and implementation means that decisions to embark on ACF rely largely on stakeholders' tacit knowledge, experience, values and preferences. Demonstrated screening yield in the same or other settings could also motivate further ACF implementation, although the number of cases detected may not be a relevant public health impact measure in itself. The above-mentioned WHO guidelines use the term 'indirect evidence'.[1] Strong 'direct evidence' exists that early diagnosis and correct TB treatment reduce morbidity, mortality and transmission. This can constitute, for example, 'indirect evidence' for ACF, based on the logic that if ACF leads to early detection and treatment, then it should also lead to better health outcomes and less transmission, despite the lack of trial data demonstrating a direct health impact of ACF. Similar reasoning can be used to describe potential harms associated with ACF, for example, based on modelling of the risk of false positive TB diagnoses in different epidemiological contexts. 'Indirect evidence', tacit knowledge, experience, values and preferences are likely to be important antecedents for national policy.

### Components

The components of ACF policy, and of the system which ACF is part of, are complex. The articles reported on the prerequisites of the given system, its resources,[46] capacity[30 42] and tools.[90] However, they also elaborated on the need for integrating ACF into health systems,[41 60] as well as educating and engaging healthcare providers,[5 21] communities,[12 29 39 44] screening participants[35] and TB patients and their families[25 34 35] for ACF implementation to succeed. All components seem indispensable for successful ACF policy development and implementation. Yet, in real-life settings, rarely all components are in place.

Which ACF components are essential and which are non-essential? The 2013 WHO guidelines on systematic screening list six key principles for screening for active TB, the first principle being: 'Before screening is initiated, high-quality TB diagnosis, treatment, care, management and support for patients should be in place, and there should be the capacity to scale these up further to match the anticipated rise in case detection that may occur as a result of screening'.[1] Yet, countries might not even completely adhere to this very first principle, for example, due to a lack of resources, before a country embarks on ACF policy development and implementation.

### Influencing factors
#### Scarce research on ACF policy development
Our findings indicate a paucity of research focusing specifically on ACF policy development. The only two factors described— politics and laws—were mentioned by the same author in two different articles and concerned migrant TB screening in a low-incidence country.[46 47]

Even though not documented as such in the literature, data on the epidemiological situation of a country and local evidence regarding impact are assumed to be

key factors influencing ACF policy development.[1] These types of data and evidence are needed to identify high-risk groups[90] and devise appropriate screening algorithms. An assessment of the epidemiological situation might be based on data from regularly collected data sets or prevalence surveys, and WHO has developed a tool to help countries estimate screening yield and costs based on such data to identify appropriate risk groups and approaches.[91]

Interestingly, the evolution of the funding landscape for ACF, for example, through TB REACH, has not been explicitly mentioned as an influencing factor for ACF policy development or implementation, though it has been a likely game-changer. TB REACH is a case-finding initiative funded largely by Global Affairs Canada and coordinated by the Stop TB Partnership.[92] The initiative has both funded ACF activities and raised awareness about the importance of ACF. It has contributed to the local evidence base, while the uptake of the evidence generated by TB REACH may be considered but not mentioned in the studies included in this review. Four articles mentioned TB REACH as funder of their research, with the disclaimer that the initiative had no role in study design, data collection and analysis.[16 42 60 61] Other publications acknowledged its role in implementing ACF when describing the study methods, for example, that TB REACH funded door-to-door screening[39] or the implementation of different ACF models[62] and that the initiative upgraded an ACF strategy by introducing new diagnostics which still have limited availability in many settings (eg, Xpert MTB/RIF).[36] It is important to reflect on and document the influence that TB REACH, other international funding mechanisms and technical agencies have on ACF policy development and implementation.

### ACF policy and social determinants of health
Many barriers and facilitators for implementing ACF policies lie at the level of communities and individuals. These factors include social determinants such as the educational level,[14 35] employment status[27 63] and access to care[64] of the target population. These factors can limit an individual's or a community's participation in ACF. ACF policies cannot be put in practice successfully if implementation barriers prevail. However, rather than talking about 'missing cases' and communities that are 'hard to reach', it may be useful to consider if health services are missing or hard to reach instead.

Individuals or communities may be ACF target groups precisely because health services are missing or hard to reach. As such, ACF could be a way to by-pass access barriers, for example, by using mobile units. Yet, even when designing services to be more accessible, target populations may still not come forward, take the next step of referral to complete the diagnostic pathway or adhere to treatment due to socioeconomic barriers.

We argue that ACF polices thus must be designed and implemented with careful consideration of social determinants in a given context to avoid possible harm.

Whitehead and Dahlgren[93] described how interventions designed to help vulnerable populations may be 'implemented in such a way as to stigmatize the very people the programme was designed to help and, in so doing, push them to avoid the help on offer.'

### A comprehensive view on ACF policies
A comprehensive approach in developing and implementing ACF policies is required that includes health education as an important component to ACF policies, and that ensures collaboration between key stakeholders, both between patients and providers, and between providers and local health authorities, between agencies, and across sectors and disease programmes.

Although this review focused on the factors that influence ACF policy and implementation, it is important to also reflect on how ACF policy influences its context in return, for example, not only factors at the level of the community and the individual, but at the level of the health system and beyond; How may the ACF policy increase or decrease stigma in a community? How may ACF policy influence the human and financial resources available in a health system? Only then may we see the real impact of ACF and be able to make equitable decisions in policy development and implementation processes.

### Future research
This scoping review demonstrated that while we know much about facilitators and barriers for ACF policy implementation, we know less about *how* to strengthen those facilitators and *how* to overcome those barriers. A major knowledge gap remains when it comes to understanding which contextual factors influence ACF policy development.

### Limitations
Our scoping review has some limitations. To increase the feasibility, we limited the search to MEDLINE, Web of Science, The Cochrane Database of Systematic Reviews and the WHO Library. We did not do a critical appraisal of the individual sources of evidence or within sources of evidence, nor did we describe sources of funding for the included sources of evidence. Such appraisals and descriptions would have allowed us to discover the scarcity of research and to characterise the quality of and gaps in the evidence base and enable the development of robust recommendations. We leave these questions for future systematic reviews to address.

We acknowledge that we may have been able to provide a more in-depth overview of antecedents, components and influencing factors for ACF policy development and implementation by including additional databases, searching grey literature and references of included studies, and contacting authors for more information, which could be included in a future systematic review. However, we believe that our review still adds value to the current body of knowledge on ACF, by providing collated and comprehensive insights into the peer-reviewed

scientific literature. We also limited the inclusion to studies written in English, which may have resulted in the exclusion of eligible studies in other languages, for example, Spanish. Furthermore, the results from this scoping review are only up to date as of 31 January 2018. The data were extracted by one reviewer only. However, the data are likely valid, as a pilot-test was conducted prior to embarking on data extraction with members of the study team. A second reviewer verified a random selection of the data.

Another limitation was that the included articles often did not distinguish between the implementation of an ACF policy versus the implementation of an ACF programme or intervention. This is likely due to inconsistent use of the terms in the literature. As such, our results are likely applicable to both. Furthermore, the reporting of ACF policy development and implementation varied in their completeness across the included articles, and as such, our data are limited by the details described in the literature, for example, most papers described steps to implementation, but did not provide details on non-response or unsuccessful practices. There is a risk of publication bias.

## CONCLUSIONS

We identified some main themes regarding the antecedents, components and influencing factors for ACF policy development and implementation. However, evidence remains scarce especially concerning policy development. Research on ACF is required to understand, inform and improve policy development and implementation.

**Acknowledgements** OB is funded by the EU-Horizon 2020-funded IMPACT-TB project (grant 733174). The authors thank Carl Gornitzki, medical librarian at Karolinska Institutet, for supporting the development of the search strategy, and Gabriella Ekman, writing instructor at Karolinska Institutet's University Library for her valuable feedback in writing this manuscript.

**Contributors** OB, KL, MC and KV conceived the study. OB developed the search strategy together with a medical librarian from Karolinska Institutet. OB and KV screened titles and abstracts, and then full-text articles. OB extracted, charted, analysed and interpreted the data. KV verified the data extraction by charting data from a random sample of studies for comparison. OB wrote the first draft of the manuscript. KL, MC and KV were consulted at various stages of the scoping review to provide input on the search strategy, data extraction, charting, analysis and the interpretation of the results, and provided comments on the manuscript. All authors read and approved the final manuscript.

**Funding** This work was supported by the EU-Horizon 2020-funded IMPACT-TB project (grant 733174).

**Competing interests** None declared.

**Patient consent for publication** Not required.

**Provenance and peer review** Not commissioned; externally peer reviewed.

**Data availability statement** All data relevant to the study are included in the article or uploaded as supplementary information.

**ORCID iD**
Olivia Biermann http://orcid.org/0000-0002-5978-0211

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
