## [Reviewer comments · BMJ Open]

ARTICLE DETAILS

TITLE (PROVISIONAL)	Factors influencing active tuberculosis case finding policy development and implementation: a scoping review
AUTHORS	Biermann, Olivia; Lönnroth, Knut; Caws, Maxine; Viney, Kerri

VERSION 1 - REVIEW

REVIEWER	Robert Makombe Senior TB Technical Advisor FHI 360 South Africa
REVIEW RETURNED	15-Jun-2019

GENERAL COMMENTS	While the key questions and objective are clearly defined, and the rationale for the scoping review implicit therein, it would be beneficial to readers for the article to be more explicit on why this approach for research synthesis was taken. Under conclusions, the scarcity of the evidence may be a result of the limited database used, which could have excluded some relevant studies. Nevertheless, the recommendation for more original research to improve understanding and policy development and implementation is very valid. The paper presents several limitations, including the possibility that some relevant studies could have been missed due to database selection. It would be useful to mention/discuss the lack of a critical appraisal of the quality of the studies reviewed, to identify gaps in the evidence base (and not just scarcity of research) and to facilitate the development of robust recommendations. All in all, this paper adds to the body of evidence around ACF. It presents and clarifies key concepts/definitions, identifies the important factors and makes clear the need for more and original research on ACF policy development and implementation.
---

REVIEWER	Jamie Rudman London School of Hygiene & Tropical Medicine
REVIEW RETURNED	02-Jul-2019

GENERAL COMMENTS	Generally minor revisions needed, but consider also restructuring the results and discussion section to align more with the abridged data charting table. This table is concise and follows a logical framework, whereas the narrative in the results and discussion sections is less structured, and some of the insights in the discussion aren't clearly linked to articles. See comments in the word document attached to this review. Factors influencing active tuberculosis case finding policy development and implementation: a scoping review Biermann, O. et. al, 2019 BMJ Open Summary of opinion  - Interesting, important, timely question/enquiry? An important question given that high burden countries are factoring ACF interventions in to strategic plans to meet SDG 2030 targets for TB elimination. This review makes the important point that evidence of successful approaches to ACF policy development and implementation is scarce. - Level of detail, clarity, explanations Overall the review is clear and concise, with results presented in a logical order set out in the research question (antecedents, components and influencing factors). Some rewording is needed to make key points clearer. Suggestions have been made below, noting the page and line of the proposed correction. The review doesn't offer much insight in to the epidemiological and programmatic considerations for ACF policy development and implementation e.g. diagnostic sensitivity and specificity, challenges in estimating prevalence of TB in the population(s) being screened, and the issue of false positive diagnoses (the latter mentioned in the discussion section). These considerations could feed in to 'health system level' and 'individual and community' subsections e.g. availability of diagnostic tests [of sufficient sensitivity and specificity] Recommend clarifying throughout the results section which studies are based on TB contact screening and which are based on migrant screening programs, as this isn't always explicit. Potential contradiction which needs clarifying: having a growing evidence base for ACF (page 15, L368), and there not being enough evidence on the benefits and epidemiological impact and cost-effectiveness of ACF (page 16, L380). Clarify what the evidence base is informing and where it is limited. This review highlights the contribution of TB REACH to developing national ACF policies – recommend giving an overview of what these policies are- given that they are possibly the most robust context-specific documentation available on ACF policy- and to use these as examples to inform the discussion.  - Final recommendation Overall a methodical review which follows PRISMA guidelines closely and presents results of the search strategy clearly and in a logical order. The review would be strengthened by providing more specific and more detailed examples of antecedents, components and influencing
---

	factors from the included articles; or at least discuss one or two examples/case studies of ACF implementation and refer to these at multiple points to reinforce messaging in the text (where feasible). Insights provided under the heading 'ACF policy and social determinants of health' although potentially very important considerations for the success of ACF policy and implementation, don't hold much weight without supporting evidence from literature. Important to state that this is speculative. Moreover, whilst components and influencing factors are clearly summarised here, the rationale for their proposed impact on ACF isn't contextualised often enough. The example of UK migrant TB screening is on page 11 is well placed, and the review would benefit from more 'cases in point' like this. Abstract - Abstract accurate, balanced and complete? Yes – but should mention the position of social determinants in the ACF policy landscape, as these are addressed as important influencing factors in the discussion section Background Minor P2, L26-27: refer to WHO's 'Systematic screening for active tuberculosis – principles and recommendations' handbook. P2, L33: start month of search not included here P2, L43: useful to provide definition of 'TB contact' e.g. targeted screening of household contacts of TB cases, if included in search criteria. P2, L47: consider rephrasing to make link more explicit, e.g. 'factors which influenced investment of government/ministries of health/donors, in/implementation of ACF'. P2, L47-48: how many articles identified refer to WHO guidelines? Need to make more explicit the link between WHO emphasis and interest in ACF in specific settings P3, L49-50: 'key components of [successful/sustained] ACF implementation include health system capacity...' to distinguish between ACF mechanism integrated in to health system vs. as part of operational research/other study. P4, L84: reorder: 'combination of an underdiagnosis of cases and underreporting of cases which have been detected' or similar. P4, L86: expand on 'economic hardship' i.e. 'for TB patients and TB affected households' or similar; also correct to 'sustained transmission'. P4, L91-92: confirm that global trends indicate highest incidence/prevalence in adults of working age. Could be that burden of TB is with elderly in some settings. P4, L93: ACF leading to an increase in case notifications. Useful to clarify definition of implementation, to distinguish ACF in context of operational research from ACF implemented routinely within TB program. P4, L97-98: referring to harm to individuals through providing TB treatment to individuals without TB? Stigma and discrimination arising from identifying cases in community that would otherwise go unnoticed? Need to clarify what is classed as harmful here. P4, L101-102: consider rephrasing to make clear that it's important to have ACF integrated in to (primary) health care system, and not delivered separately.
--	---

	P5, L103-104: consider rephrasing to make clearer that the aim of evidence generation is to identify specific approaches to ACF which are context-appropriate, cost-effective and efficient, e.g. 'the strength of evidence to support alternative approaches to ACF is low, with few head-to-head ACF trial results available to inform best practice'. P5, L107-109: useful to have specific examples of national policies here, citing relevant papers. Review methods and data sources  - Repeatability based on description? Yes, this could be readily repeated based on methods provided. P6, L141: detection/identification of active TB disease?  - Study design appropriate to research question? Yes – scoping review appropriate for this question.  - Follows PRISMA ScR checklist for reporting? Yes – checklist included in supplementary materials Findings and interpretation  - Outcomes clearly defined? Yes: total number of titles and abstracts screened, and total fulfilling eligibility criteria are stated. Would be good to summarise the diversity of ACF policies and characteristics identified here, and/or provide some examples which span the breadth of the policies and their components. First year of search based on first year for which any ACF literature available? Or chosen on other grounds?  - Statistic used appropriately and described in the study? N/A  - Results address the research question or objective? Clear stated the proportion of total articles which focused on ACF for TB contacts and migrants, but not clear what target population (if any) the remaining 43% articles addressed. Social determinants feature in the discussion but are not mentioned in the summary of results on page 2. Objectives: explore antecedents, components and influencing factors. Each of the three aspects of the research question are addressed under clear subheadings and presented in a logical order. Antecedents P9, L213-214: include specific example of HIV equivalent? P9, L222-223: provide specific conditionalities/criteria, making references to articles. P10, L225-226: wording not clear- referring to operational research to inform implementation strategy? P10, L229-230: evaluations of policy, or epidemiological context? Components: The components section needs an introductory paragraph which summarizes the main building blocks of ACF. P10, L233-234: either include definition of ACF and summary of building blocks earlier in the paper and/or refer to relevant paper(s), so a reader can easily gain more context where needed. P10, L236-237: number needed to screen to reach specific notification targets, or to detect and treat one 'missing' case? P10, L242-243: state if incentives are for TB patients and TB affected households. P11, L251: rephrase: existing systems and services
--	---

	P11, L252: this needs more clarity; again, some context/country-specific examples from the literature would help to clarify what is meant by 'vertical' and 'stand-alone' structures in the context of ACF. Influencing factors: P11, L267-268: granted the results are summarised in table 2 and detailed under subheadings 'health system level' and 'individual and community', it's important to have a narrative running through the text with some (condensed) examples of facilitators and barriers. P11, L278: remove question mark. P12, L288-289: [lack of] availability of health and social services? Or refers to the need to invest in these areas? Are there examples of efforts to scale up these services and networks – if so perhaps include here. Categories of barriers and facilitators is reasonably exhaustive. Not always clear if there are examples from articles of facilitators being invested in or implemented (even if in operational research context). E.g. 'availability of molecular diagnostic tests such as Xpert MTB/RIF' doesn't indicate if these tools have been successfully implemented for ACF. P13, L303: consider rephrasing, e.g. '16% of articles mentioned staff overburden and resulting staff absenteeism as an inhibitor of successful ACF implementation'. P13, L308: consider addition to this sentence: 'collaboration between community workers, health centre staff and laboratory technicians [in delivering diagnostic services as part of an ACF campaign]' or other appropriate end-point. P13, L311-312. Need clarification on the provider and beneficiary, e.g. '[healthcare providers] being non-judgemental and supportive [of patients and their families/households, during diagnosis and follow up]'. P13, L313: clarify whether health education is for patients, TB-affected households, and/or communities more widely. P13, L322-323: define contact e.g. 'of an index TB case'. Or does this refer to community contact in the wider sense? P14, 324-325. Needs rewording e.g. 'Issue of fear, either of TB disease, of the process of [community] ACF, or the wider implications of being identified as a TB case e.g. for migrant screening programmes, where being identified as a TB case has in some instances resulted in deportation (citation). P14, L326-327: consider rephrasing 'articles report [lack of] trust of patients and families/households, as inhibitors of patients receiving early diagnosis...' P14, L336: 'receptiveness [to] community workers' advice'. - Discussions and conclusions justified by results? Yes for the most part – see comments below P15, L358: consider rephrasing to make the link to WHO publication explicit: 'which may be a result of programmes implementing WHO guidelines on systematic screening' or similar emphasis on guidelines as a catalyst for policy development and publications. P15, L368-372: good to include this in the methods section instead of or in addition to here. P16, L381: need to clarify here that there is a lack of evidence on the benefits of ACF strategies, acknowledging that there are multiple approaches/building blocks.
--	--

	Contradiction about having a growing evidence base, and there not being enough evidence on the benefits of ACF. Clarify what the evidence base is informing and its limitations. P16, L387-388: needs rephrasing for clarity. P17, L405-408: 'need for integration, education and engagement' is a very general statement: outline what ACF is being integrated in to (e.g. health system) and what engagement is needed and with who (community, healthcare providers etc). P17, L410: rephrase 'which ACF components are essential and which are non-essential' or similar. P18, L416: rephrase for clarity P18, L425-426: this assumption is from WHO guidelines on systematic screening? P18, L450-452: not clear what this means – social determinants have a compound negative impact on the ability of individuals and communities to participate in ACF? P19, L453-455: good insight, has this angle been discussed in articles included in this review? If articles have explored this, what is meant by 'hard to reach'. Include specific examples. P19, L456-457: unclear – individuals and communities may be targeted for ACF because health services are missing or hard to reach, as opposed to the individuals/communities being hard to reach. P19, L462 to P20, L479: these seem more general discussion points and could be presented under a separate heading. What is the evidence base to support the statement 'ACF tends to be designed as a short cut to reaching the poor'? Is this the key message here? – An ACF campaign has an impact on the community and health system which can be both positive and negative. Implementing ACF may result in an increase in the investment in human resource capacity, and thus be beneficial not just to the target population but to the wider health system. If so, needs clarification. Clarity  - Results presented clearly? Results follow the logical order set out in the research question and objectives, with a concise summary of the results of the literature search provided at the start. Suggest separating facilitators and barriers under subheadings to make the distinction clearer. The abridged data charting tables provides a clear summary of key findings from the articles, categorised in to antecedents, components and influencing factors. The text lacks this clarity at points - recommend the main text in the results follows more closely the order and specific points included in the table.  - Study limitations discussed accurately? Yes, but justification of choice to not expand search to grey literature and operational research studies is needed. Given the emphasis on social determinants in the discussion section, context-specific operational findings and even anthropological literature could provide insight on barriers to ACF implementation. Key limitation stated here is that articles didn't distinguish between implementation of policy vs. implementation of an
--	---

	ACF program or intervention. No clear strategy for mitigating this provided here.  - Standard of written English acceptable for publication? Yes, for the most part – suggestions for corrections to phrasing given above. Motivation/research question  - Clear definition of study objective? Yes. Studies on the effectiveness, yield, accuracy and impact of ACF were only selected if they included description of how this evidence/insight has or could influence policy.  - Research ethics addressed? Review framing  - Supplementary reporting complete (PRISMA checklist)? Yes. PRISMA flow diagram in Figure 1 clearly indicates how final number of studies arrived at for inclusion in data-charting step.
--	---

VERSION 1 – AUTHOR RESPONSE

#	Reviewers' comments	Authors' answers
	Overall comments	
1	It would be beneficial to readers for the article to be more explicit on why this approach for research synthesis was taken. (Reviewer 1, Robert Makombe)	We added on P6, L136-137: "The scoping review methodology was deemed appropriate given the breadth of the question."
2	It would be useful to mention/ discuss the lack of a critical appraisal of the quality of the studies reviewed to identify gaps in the evidence base (and not just scarcity of research) and to facilitate the development of robust recommendations. (Reviewer 1, Robert Makombe)	We added on P23-24, L573-583: "Such appraisals and descriptions would have allowed us to not only discover the scarcity of research, but to characterize the quality of and gaps in the evidence base and enable the development of robust recommendations. We leave these questions for future systematic reviews to address. We acknowledge that we may have been able to provide a more in-depth overview of antecedents, components and influencing factors for ACF policy development and implementation by including additional databases, searching grey literature and references of included studies, and contacting authors for more information, which could be included in a future systematic review. However, we believe that our review still adds value to the current body of knowledge on ACF, by providing collated and comprehensive insights into the peerreviewed scientific literature.

3	Restructuring the results and discussion section to align more with the abridged data charting table. This table is concise and follows a logical framework, whereas the narrative in the results and discussion sections is less structured, and some of the insights in the discussion aren't clearly linked to articles. (Reviewer 2, Jamie Rudman)	In contrast, reviewer 1 was of the opinion that “the review is clear and concise, with results presented in a logical order set out in the research question (antecedents, components and influencing factors)” and that “Each of the three aspects of the research question are addressed under clear subheadings and presented in a logical order.” We did critically review and compare the structure of the abridged data charting table with the structures of the results and discussion sections. Finally, we found that they were in line and therefore opted for leaving the structure as is. We have, however, added references in the discussion session to clearly link the insights to articles, as recommended.
	Summary of opinion	

#	Reviewers' comments	Authors' answers
4	The review doesn't offer much insight in to the epidemiological and programmatic considerations for ACF policy development and implementation e.g. diagnostic sensitivity and specificity, challenges in estimating prevalence of TB in the population(s) being screened, and the issue of false positive diagnoses (the latter mentioned in the discussion section). These considerations could feed in to 'health system level' and 'individual and community' subsections e.g. availability of diagnostic tests [of sufficient sensitivity and specificity].	The review does not offer much insight into the epidemiological and programmatic considerations as the articles included in this review did not provide such insights. We therefore did not mention these factors under “health system level” and “individual and community” in the results section. However, we mention on P8, L185-187 other studies that will add on this: “This study is related to a qualitative study of experts and a survey of National TB Programme managers combined with a policy document review on the topic of ACF policy development and implementation which are currently under way to harness tacit knowledge on the topic.” In addition, in the discussion section, we discuss epidemiology as an important influencing factor on P20, L501-503: “Even though not documented as such in the literature, data on the epidemiological situation of a country and local evidence regarding impact are assumed to be key factors influencing ACF policy development [1]. These types of data and evidence are needed to identify high-risk groups [62] and devise appropriate screening algorithms.” We also added on P14, L348: “Availability of diagnostic tests of sufficient sensitivity and specificity”, on P18, L427429: “Most studies were published after 2010, which may be a result of programmes implementing WHO

		guidelines on systematic screening [1] as well as the growing number of prevalence surveys in different settings [35].” And on P21 L519 we added: “the initiative upgraded an ACF strategy by introducing new diagnostics which still have limited availability in many settings (e.g. Xpert MTB/RIF®) [51].”
5	Recommend clarifying throughout the results section which studies are based on TB contact screening and which are based on migrant screening programs, as this isn’t always explicit.	To keep the text flow and given that there are not only contacts and migrants, but also different communities, high-risk groups, homeless, asylum seekers, refugees, people living with HIV, urban poor and other groups, we have added an additional reference to Table 1 for details: “What all included studies have in common is the goal of finding ‘missing’ TB cases through ACF, while the approaches vary greatly; from contact investigation to screening homeless shelters (Table 1).” (P9, L198-200)
6	Potential contradiction which needs clarifying: having a growing evidence base for ACF	On P18, L438, we deleted: “The evidence base is not only growing but diverse.” In addition, we clarified on P18, L450: Despite the growing number of studies

#	Reviewers’ comments	Authors’ answers
	(page 15, L368), and there not being enough evidence on the benefits and epidemiological impact and cost-effectiveness of ACF (page 16, L380). Clarify what the evidence base is informing and where it is limited.	(e.g. operational studies), there is still a relative lack of evidence about the health benefits for individuals, the epidemiological impact and cost-effectiveness of different ACF approaches.
7	This review highlights the contribution of TB REACH to developing national ACF policies – recommend giving an overview of what these policies are- given that they are possibly the most robust context specific documentation available on ACF policy- and to use these as examples to inform the discussion.	We highlight TB REACH’s contribution to the evidence base on P21, L513-515: TB REACH “has contributed to the local evidence base, while the uptake of the evidence generated by TB REACH may be considered but not mentioned in the studies included in this review.” As the review does not highlight the contribution of TB REACH to developing national policies, we are unable to provide an overview of TB REACH-influenced policies. However, this would indeed be an interesting questions for a future study.

8	The review would be strengthened by providing more specific and more detailed examples of antecedents, components and influencing factors from the included articles; or at least discuss one or two examples/case studies of ACF implementation and refer to these at multiple points to reinforce messaging in the text (where feasible).	We included an example of a study examining ACF implementation on P9, L198-209: “An example of a study included in this review is a qualitative study by Ayakaka et al., which explored influencing factors for TB contact investigation in Kampala, Uganda [29]. The stakeholders who were interviewed described key barriers for ACF as being limited knowledge about TB among contacts, stigma, mistrust of health centre staff among index patients and contacts, and high travel costs for lay health workers and contacts. At the same time, key facilitators for ACF comprised personalized and enabling services provided by lay health workers. To overcome barriers and strengthen facilitators, the researchers identified education, incentivization and restructuring of the service environment as relevant interventions.”
9	Insights provided under the heading ‘ACF policy and social determinants of health’ although potentially very important considerations for the success of ACF policy and implementation, don’t hold much weight without supporting evidence from	On P22, L539, we clarified when content was speculative by adding “We argue...”

#	Reviewers’ comments	Authors’ answers
	literature. Important to state that this is speculative. Moreover, whilst components and influencing factors are clearly summarised here, the rationale for their proposed impact on ACF isn’t contextualised often enough. The example of UK migrant TB screening is on page 11 is well placed, and the review would benefit from more ‘cases in point’ like this.	On P15, L367-370, we added an example demonstrating how an influencing factor impacted ACF results: “Moreover, Akkerman et al. described the positive impact on diagnostic accuracy and the proportion of follow-up medical interventions when there is close collaboration with a limited number of experienced, high-volume chest radiograph readers when screening asylum seekers and other high-risk groups [77].”
	Abstract	

10	Yes – but should mention the position of social determinants in the ACF policy landscape, as these are addressed as important influencing factors in the discussion section.	We added on P2, L46-48: “Fifty-nine percent of studies reported influencing factors for ACF implementation; mostly linked to the health system (e.g. resources) and the community/individual (e.g. stigma social determinants of health).”
	Background	
11	P2, L26-27: refer to WHO’s ‘Systematic screening for active tuberculosis – principles and recommendations’ handbook.	While we refer to the WHO guidelines in the background section of the manuscript amongst others, we suggest sticking to “The WHO emphasized the importance...” in the abstract on P2, L26-27 due to the word limit.
12	P2, L33: start month of search not included here.	On P2, L35 we now state the start month of the search: January.
13	P2, L43: useful to provide definition of ‘TB contact’ e.g. targeted screening of household contacts of TB cases, if included in search criteria.	We did not add a definition of “TB contact” here, as “TB contact” is included as follows in the search strategy: “contact tracing OR contact examination OR contact screening OR contact investigation”. In addition, the word limit made it difficult to elaborate on the definition of a TB contact in the abstract. However, we defined “contact” on P16, L387-389: “The consequences could be a contact’s the avoidant behaviour of a contact of an index TB case.”
14	P2, L47: consider rephrasing to make link more explicit, e.g. ‘factors which influenced investment of government/ministries of health/ donors, in/implementation of ACF’.	As we explicitly referred to “policy development” in the search strategy, we did not rephrase it in the abstract. Nevertheless, we elaborated on it as suggested on P13, L314: “Policy development comprises the investment of governments and donors in ACF.”

#	Reviewers’ comments	Authors’ answers
15	P2, L47-48: how many articles identified refer to WHO guidelines? Need to make more explicit the link between WHO emphasis and interest in ACF in specific settings	On P3, L50 we clarified that six articles referred to WHO guidelines. However, due to word limitations in the abstract, we left the formulation more general as WHO’s recommendations from 1976 rather influenced ACF in high-income countries, and the 2013 guidelines rather impacted ACF in low- and middle-income countries.

16	P3, L49-50: 'key components of [successful/sustained] ACF implementation include health system capacity...' to distinguish between ACF mechanism integrated into health system vs. as part of operational research/other study.	We added on P3, L52: "Key components of successful ACF implementation include health system capacity, mechanisms for integration, education and collaboration for ACF."
1§7	P4, L84: reorder: 'combination of an underdiagnosis of cases and underreporting of cases which have been detected' or similar.	We reordered on P4, L89-90: "a combination of an underdiagnosis of cases and underreporting of cases who have been detected TB cases and underdiagnosis [3]"
18	P4, L86: expand on 'economic hardship' i.e. 'for TB patients and TB affected households' or similar; also correct to 'sustained transmission'.	We expanded on P4, L91-92: "and economic hardship for TB patients and TB-affected households, and sustaining ed transmission"
19	P4, L91-92: confirm that global trends indicate highest incidence/ prevalence in adults of working age. Could be that burden of TB is with elderly in some settings.	We confirm. Globally, and in most settings, the highest absolute number of incident TB cases is indeed among people aged 18-64, but it depends on both epidemiological and demographic situation. Incidence rate is in fact highest among young children and elderly. We have therefore written that TB "often" affects the most economically productive members of a society (P4, L95-98).
20	P4, L93: Useful to clarify definition of implementation, to distinguish ACF in context of operational research from ACF implemented routinely within TB program.	As we did not distinguish between ACF in the context of operational research vs. ACF in the context of routine TB program implementation, we did not add this definition. However, this would be a useful distinction and interesting questions for a future study.
21	P4, L97-98: referring to harm to individuals through providing TB treatment to individuals without TB? Stigma	We clarified on P5, L104-107: "It can also cause harm to individuals, e.g. by increasing the risk of a false positive diagnosis and providing TB treatment to individuals without TB, or increased stigma and

#	Reviewers' comments	Authors' answers
	and discrimination arising from identifying cases in community that would otherwise go unnoticed? Need to clarify what is classed as harmful here.	discrimination arising from uncovering TB, which need to be weighed against the benefit of identifying cases in the community that would otherwise go untreated [1,2]."

22	P4, L101-102: consider rephrasing to make clear that it's important to have ACF integrated in to (primary) health care system, and not delivered separately.	We rephrased on P5, L110-111: "implement outreach screening through ACF in a given context as a synergistic way, i.e. integrated into the health care system, and not delivered separately in a parallel system."
23	P5, L103-104: consider rephrasing to make clearer that the aim of evidence generation is to identify specific approaches to ACF which are context-appropriate, costeffective and efficient, e.g. 'the strength of evidence to support alternative approaches to ACF is low, with few head-to-head ACF trial results available to inform best practice'.	We rephrased on P5, L113-115: "There strength of evidence to support alternative is weak evidence for choosing the best ACF approaches is low, and with few head-to-head trials results being available to inform practice."
24	P5, L107-109: useful to have specific examples of national policies here, citing relevant papers.	We added an example on P5-6, L120-124: "For example, Vietnam's National Strategic Plan 20152020 includes ACF in remote and congregate settings, and in high-risk groups including contacts, prisoners, miners, the elderly, homeless people, young adults, migrants, factory workers, communities, people living with Human Immunodeficiency Virus (HIV), multidrug-resistant patients, young males, smokers and intravenous drug users [11]."
	Review methods	
25	P6, L141: detection/identification of active TB disease?	We added on P7, L159: "Eligible studies focused on the detection of active TB disease."
	Findings and interpretation	
26	Would be good to summarise the diversity of ACF policies and characteristics identified here, and/or provide some examples which span the breadth of the policies and their components.	We added on P9, L198-200: "What all included studies have in common is the goal of finding 'missing' TB cases through ACF, while the approaches vary greatly; from contact investigation to screening people in homeless shelters." Further, in addition to the examples on screening migrants (P13, L316-321), we added examples on contact tracing (P9,

#	Reviewers' comments	Authors' answers
		L201-209) and screening of asylum seekers and other high-risk groups (P15, L367-370).

27	Clear stated the proportion of total articles which focused on ACF for TB contacts and migrants, but not clear what target population (if any) the remaining 43% articles addressed.	We clarified on P9-10, L220-223: “The most frequent ACF target groups were contacts (25%, n=18/73) and migrants from high-incidence countries (32%, n=23/73), while . Fourteen 14% percent (n=10/73) of the articles focused on screening of defined communities. The remaining studies focused on different target groups which are specified in Table 1.”
28	P9, L213-214: include specific example of HIV equivalent?	We included on P10 L244-246: “similar to community-based testing services that have increased access to HIV testing and care Human Immunodeficiency Virus (HIV) programmes [16,17].”
29	P9, L222-223: provide specific conditionalities/criteria, making references to articles.	We provided more information and a reference on P11, L255-261: “Conditional recommendations are those for which the benefits are expected to outweigh the risks, while the trade-offs, such as costeffectiveness, feasibility or affordability may be uncertain. For example, WHO has made a conditional recommendation (very low-quality evidence) for screening certain clinical risk groups in settings where the TB prevalence is over 100 per 100,000 population (while not recommending such screening in countries with a prevalence of less than 100 per 100,000 population) [1].”
30	P10, L225-226: wording not clear-referring to operational research to inform implementation strategy?	We clarified on P11, L263-265: “The guidelines provide principles for how to cautiously experiment with test and expand an approach that had been largely missing in previous global TB strategies”
31	P10, L229-230: evaluations of policy, or epidemiological context?	We added on P11 L267-269: “Moreover, the guidelines emphasize the importance of basing ACF policy on local epidemiology and to perform careful evaluations of ACF implementation, yield and impact to improve the ACF evidence base”
32	The components section needs an introductory paragraph which summarizes the main building blocks of ACF.	We added an introductory paragraph on P11-12, L273-278: “According to WHO [1], the main components of ACF include 1) having high-quality TB diagnosis, treatment, care, management and support for patients, as well as capacity to scale these up (if needed) before screening is initiated; 2) prioritizing risk groups; 3) choosing a suitable screening algorithm; 4) following established ethical principles; 5) optimizing synergies with the delivery

#	Reviewers' comments	Authors' answers
---	---------------------	------------------

		of other health and social services; and 6) monitoring and re-assessing ACF approaches.”
33	P10, L233-234: either include definition of ACF and summary of building blocks earlier in the paper and/or refer to relevant paper(s), so a reader can easily gain more context where needed.	A definition of ACF is included on P4, L78-82. We added a summary on P11-12, L273-278. Moreover, we rephrased on P12, L279-281: “The building blocks of components of ACF, as well as those of the wider health system that ACF depends on, are multifaceted and well-described in the literature, and so are the components of the wider health system that ACF depends on.”
34	P10, L236-237: number needed to screen to reach specific notification targets, or to detect and treat one ‘missing’ case?	We clarified on P12 L284-285: “the number needed to screen to detect a case of TB and the costeffectiveness of ACF”
35	P10, L242-243: state if incentives are for TB patients and TB affected households.	We clarified on P12, L289-291: “Incentives are sometimes deemed necessary, e.g. for health workers [5,32], laboratory technicians [73], screening participants [11,36,40,42,43,67] and TB patients [32,51,67]. [23,28,30,31,32], e.g. for identifying cases [31], and in the form of cell phone airtime, food or money [23].” In addition, we also clarified the incentives and their recipients in the abridged data charting table (Additional file 3).
36	P11, L251: rephrase: existing systems and services	We rephrased on P12-13: 301-303: “and to integrate ACF within available existing systems [18,44] and services to avoid vertical and stand-along implementation structures [26,38,44].”
37	P11, L252: this needs more clarity; again, some context/country-specific examples from the literature would help to clarify what is meant by ‘vertical’ and ‘standalone’ structures in the context of ACF.	We deleted on P13: 301-302: “and to integrate ACF within existing systems [18,44] and services to avoid vertical and stand-along implementation structures [26,38,44].” Examples have been included (see comment 26 above).
38	P11, L267-268: granted the results are summarised in table 2 and detailed under subheadings ‘health system level’ and ‘individual and community’, it’s important to have a narrative running through the text with some (condensed) examples of facilitators and barriers.	Instead of adding more (condensed) examples of facilitators and barriers throughout the text, and in order to avoid redundancies, we refer to Table 2 (P14, L325) and the summarized themes at the beginnings of the sections (P14, L335-338 and P16, L381-383).

#	Reviewers' comments	Authors' answers
39	P11, L278: remove question mark.	Question mark removed on P14, L330.
40	P12, L288-289: [lack of] availability of health and social services? Or refers to the need to invest in these areas? Are there examples of efforts to scale up these services and networks – if so perhaps include here.	We clarified on P14, L345-347: "Studies elaborated on the availability of health and social services [29,18] and laboratory networks [23] that could be used for ACF implementation." We do not have examples of efforts to scale up these services and networks based on our review of the literature but would be curious to explore those in a future study.
41	Not always clear if there are examples from articles of facilitators being invested in or implemented (even if in operational research context). E.g. 'availability of molecular diagnostic tests such as Xpert MTB/RIF' doesn't indicate if these tools have been successfully implemented for ACF.	We could only discuss in general terms the reported importance of facilitators being in place, since the studies did not reveal whether these factors already existed and were being invested in, or whether they were implemented solely for ACF.
42	P13, L303: consider rephrasing, e.g. '16% of articles mentioned staff overburden and resulting staff absenteeism as an inhibitor of successful ACF implementation'.	We rephrased on P15, L256-257: "The overburdening of staff was highlighted by 16% (n=7/43) of the articles, e.g. leading to absenteeism [52], and thus inhibiting ACF implementation."
43	P13, L308: consider addition to this sentence: 'collaboration between community workers, health centre staff and laboratory technicians [in delivering diagnostic services as part of an ACF campaign]' or other appropriate end-point.	We added on P15, L365-367: "One example is the collaboration between community workers, health centre staff and laboratory technicians in delivering diagnostic services as part of ACF [15]."
44	P13, L311-312. Need clarification on the provider and beneficiary, e.g. '[healthcare providers] being non-judgemental and supportive [of patients and their families/ households, during diagnosis and follow up]'.	We clarified on P15, L373-375: "e.g. adopting approaches including health care providers being nonjudgmental and supportive of patients and their families during diagnosis and follow-up, [53] and assuring confidentiality [24,54]."

45	P13, L313: clarify whether health education is for patients,	We clarified on P15-16, L376-378: “articles emphasized health education for the general public,
----	--	---

#	Reviewers' comments	Authors' answers
	TB-affected households, and/or communities more widely.	screening participants, as well as TB patients and their families as a necessity.”
46	P13, L322-323: define contact e.g. ‘of an index TB case’. Or does this refer to community contact in the wider sense?	We defined “contact” on P16, L386-387: “The consequences could be a contact’s the avoidant behaviour of a contact of an index TB case.”
47	P14, 324-325. Needs rewording e.g. ‘Issue of fear, either of TB disease, of the process of [community] ACF, or the wider implications of being identified as a TB case e.g. for migrant screening programmes, where being identified as a TB case has in some instances resulted in deportation (citation).	We reworded on P16, L388-392: “Issues of fear, either of Closely linked was the issue of fear, which could be related to TB itself disease [25,35,53], the ACF process of ACF [55], or the wider implications of being identified as a TB case, e.g. through migrant screening programmes, where being diagnosed with TB has in some instances resulted in deportation to deportation in some cases of migrant screening [34,43,56,57].”
48	P14, L326-327: consider rephrasing ‘articles report [lack of] trust of patients and families/households, as inhibitors of patients receiving early diagnosis...’	We rephrased on P16, L393-395: “Studies also described reported the lack of trust of patients and their families as key barrier for ACF policy implementation”
49	P14, L336: ‘receptiveness [to] community workers’ advice’.	We added “to” on P17, L404.
	Discussions and conclusions	
50	P15, L358: consider rephrasing to make the link to WHO publication explicit: ‘which may be a result of programmes implementing WHO guidelines on systematic screening’ or similar emphasis on guidelines as a catalyst for policy development and publications.	We rephrased and added on P18, L 427-429: “Most studies were published after 2010, which may be linked to the publication of the a result of programmes implementing WHO guidelines on systematic screening in 2013 [1] as well as the growing number of prevalence surveys in different settings [35].”
51	P15, L368-372: good to include this in the methods section instead of or in addition to here.	We deleted on P18, L438-439: This implies that while many lessons can be learned across different settings, some might not be generalizable. and included on P7, L167-168: “The inclusion of different ACF approaches implies that while many lessons can be learned across different settings, some might not be generalizable.”

52	P16, L381: need to clarify here that there is a lack of evidence on the benefits of ACF strategies, acknowledging that	We clarified on P18-19, L450-452: “Despite the growing number of studies (e.g. operational studies), there is still a relative lack of evidence about the health benefits for individuals, the epidemiological
----	--	--

#	Reviewers' comments	Authors' answers
	there are multiple approaches/building blocks.	impact and cost-effectiveness of different ACF approaches.“
53	Contradiction about having a growing evidence base, and there not being enough evidence on the benefits of ACF. Clarify what the evidence base is informing and its limitations.	We clarified on P18, L450-452: “Despite the growing number of studies (e.g. operational studies), there is still a relative lack of evidence about the health benefits for individuals, the epidemiological impact and cost-effectiveness of different ACF approaches.“
54	P16, L387-388: needs rephrasing for clarity.	We rephrased on P19, L457-460: However, the Sustainable Development Goals – which aim to “leave no one behind” – might have contributed to a be yet another part of the explanation for the greater interest in ACF to leave no undiagnosed TB patients behind.
55	P17, L405-408: ‘need for integration, education and engagement’ is a very general statement: outline what ACF is being integrated in to (e.g. health system) and what engagement is needed and with who (community, healthcare providers etc).	We specified on P20, L479-482: “but However, they also elaborated on the need for integratingon ACF into health systems [38,45], as well as educatingon and engagingement health care providers [5,23], communities [12,32,35,44], screening participants [67] and TB patients and their families [27,43,67] for ACF implementation to succeed.”
56	P17, L410: rephrase ‘which ACF components are essential and which are non-essential’ or similar.	We rephrased on P20, L485-486: Which ACF components are essential and which are non-essential more dispensable than others?
57	P18, L416: rephrase for clarity	We rephrased on P20, L490-93: Yet, countries might not even completely adhere to this very first principle, e.g. might not be completely adhered to, often due to a lack of resources, before a country embarks on ACF policy development and implementation.
58	P18, L425-426: this assumption is from WHO guidelines on systematic screening?	Yes. We added the reference on P20, L501-503: Even though not documented as such in the literature, data on the epidemiological situation of a country and local evidence regarding impact are assumed to be key factors influencing ACF policy development [1].

59	P18, L450-452: not clear what this means – social determinants have a compound negative impact on the ability of individuals and communities to participate in ACF?	We clarified on P22, L529-530: “These factors can are inextricably linked and downwardly additive, with the power to limit an individual's or a community's participation in ACF.”
----	---	--

#	Reviewers' comments	Authors' answers
69	P19, L453-455: good insight, has this angle been discussed in articles included in this review? If articles have explored this, what is meant by 'hard to reach'. Include specific examples.	This angle has not been discussed in the articles included in the review, but was added by our team (P22, L531-533).
70	P19, L456-457: unclear – individuals and communities may be targeted for ACF because health services are missing or hard to reach, as opposed to the individuals/communities being hard to reach.	We clarified on P22, L534-535: “Individuals or communities may be ACF target groups precisely because health services are missing or hard to reach due to the described vulnerabilities.”

71	P19, L462 to P20, L479: these seem more general discussion points and could be presented under a separate heading. What is the evidence base to support the statement 'ACF tends to be designed as a short cut to reaching the poor'? Is this the key message here? – An ACF campaign has an impact on the community and health system which can be both positive and negative. Implementing ACF may result in an increase in the investment in human resource capacity, and thus be beneficial not just to the target population but to the wider health system. If so, needs clarification.	We restructured the text on P23, L545-560: "Nevertheless, ACF tends to be designed as a short cut to reaching the poor, since addressing underlying determinants "upstream" and health systems strengthening are considered more complex and timeconsuming. Whitehead et al [70] described how interventions designed to help vulnerable populations may be 'implemented in such a way as to stigmatize the very people the programme was designed to help and, in so doing, push them to avoid the help on offer.' A comprehensive view on ACF policies In addition, a A comprehensive approach in developing and implementing ACF policies is required that includes health education as an important component to ACF policies, and that ensures collaboration between key stakeholders, not only between patients and providers, but between providers and local health authorities, between agencies, and across sectors and disease programmes. Finally, a Although this review focused on the factors that influence ACF policy and implementation, it is important to also reflect on how ACF policy influences its context in return, e.g. not only factors at the level of the community and the individual, but at the level of the health system and beyond;. How may the ACF policy increase or decrease stigma in a community? How may ACF policy influence the human and financial resources available in a health
#	Reviewers' comments	Authors' answers
		system? Only then may we see the real impact of ACF and be able to make equitable decisions in policy development and implementation processes."
	Clarity	

72	Suggest separating facilitators and barriers under subheadings to make the distinction clearer.	Instead of separating facilitators and barriers under the subheadings, we added an explanation for why we are hesitant to doing it on P13, L306-310: "Influencing factors: In the following section, we summarize influencing factors for ACF policy development and implementation. We chose to speak of "influencing factors" rather than facilitators and barriers, as one and the same factor might be a facilitator in one context and a barrier in another, e.g. health workers' satisfaction might be a powerful facilitator for ACF [39], while the lack thereof could be a strong barrier as well [46]."
	Limitations	
73	justification of choice to not expand search to grey literature and operational research studies is needed.	The main reason for not expanding the search is feasibility (P23, L569). We included any type of study as part of our inclusion criteria.
74	Key limitation stated here is that articles didn't distinguish between implementation of policy vs. implementation of an ACF program or intervention. No clear strategy for mitigating this provided here.	The lack of distinguishing between policy versus program implementation is a limitation of the articles included in the review and could thus not be mitigated. Instead, we are convinced that the results are likely applicable to both (P24-25, 591-594).

VERSION 2 – REVIEW

REVIEWER	Robert Makombe FIH 360 South Africa
REVIEW RETURNED	29-Aug-2019

GENERAL COMMENTS	The responses to the questions raised by the other reviewer have strengthened the manuscript, which adds to the body of evidence around the development and implementation of policy on active case finding for TB.
---

REVIEWER	Mr Jamie Rudman London School of Hygiene & Tropical Medicine
REVIEW RETURNED	16-Oct-2019

GENERAL COMMENTS	Authors have carefully considered all/most comments and have restructured and rephrased where needed, providing strong justifications for not making some changes. The review article is now more clear and concise, with some more case studies/examples included which help to contextualize main points. One possibility for consolidating the review's key take-home messages is to expand on conclusion section (word limit permitting), using wording consistent with main body of text. E.g. L52-53 'key components of successful ACF implementation include suitable financial backing/investment and mechanisms of integration of ACF in to health system, the capacity of the health system to provide and sustain ACF service provision in the longer term, education programs for TB affected households, communities and for health providers; and (lastly) collaboration between key actors/stakeholders'. I would say very much optional to add this level of detail here, but I've recommended this as a minor revision for authors to consider before resubmitting.
--

VERSION 2 – AUTHOR RESPONSE

	Reviewers' comments	Authors' answers
3	One possibility for consolidating the review's key take-home messages is to expand on conclusion section (word limit permitting), using wording consistent with main body of text. E.g. L52-53 'key components of successful ACF implementation include suitable financial backing/investment and mechanisms of integration of ACF in to health system, the capacity of the health system to provide and sustain ACF service provision in the longer term, education programs for TB affected households, communities and for health providers; and (lastly) collaboration between key actors/stakeholders'.	We expanded the conclusion section on P3, L54-61 to consolidate the key take-home messages. Given the broad scope of the results and the word limit, we suggest describing the concluding messages more broadly, instead of summarizing concrete results. “We identified some main themes regarding the antecedents, components and influencing factors for ACF policy development and implementation. While we know much about facilitators and barriers for ACF policy implementation, we know less about how to strengthen those facilitators and how to overcome those barriers. A major knowledge gap remains when it comes to understanding which contextual factors influence ACF policy development. However, evidence remains scarce especially concerning policy development. More original Research is required to understand, inform and improve ACF policy processes development and implementation.”